# Does Schooling Causally Impact Non-Cognitive Skills? Evidence from Elimination of Social Security Student Benefits

**Bahram Sanginabadi**

Economics Department, University of Hawaii at Manoa, Honolulu, HI 96822, USA; bahram@hawaii.edu

**Abstract:** A limited number of studies have investigated the impacts of education on non-cognitive skills, yet they offer mixed results. A few studies suggest no impact, but others report positive impacts of education on non-cognitive skills. In this paper, we apply the elimination of Social Security student benefits that took place in the United States in 1982 to study the impacts of education on non-cognitive skills, as measured by the Rotter Locus of Control Scale and Rosenberg Self-Esteem Scale. We apply eligibility for aid due to the death of father to avoid endogeneity in our analysis. Our results suggest that non-cognitive skills improve during the college education years, but the causality relationship from college education to non-cognitive skills disappears to a high extent when the prior levels of non-cognitive skills are controlled for.

**Keywords:** schooling; non-cognitive skills; self-esteem; locus of control

## 1. Introduction

The two well-known measures of non-cognitive skills, the locus of control scale, and the self-esteem scale, have important educational and professional applications. Numerous studies focus on estimating the impacts of non-cognitive skills on important life outcomes, but studies on the impacts of formal schooling on non-cognitive skills remain rare. While researchers have paid attention to the impacts of education on cognitive skills, little attention is given to non-cognitive skills. Focus has been especially centered on crystallized intelligence (Ceci 1991; Murray and Herrnstein 1994; Stelzl et al. 1995; Neal and Johnson 1996; Winship and Korenman 1997; Hansen et al. 2004; Blair et al. 2005; Cascio and Lewis 2006; Cliffordson and Gustafsson 2008; Arcidiacono et al. 2010; Carlsson et al. 2015; Gustafsson 2016; Jonsson et al. 2017).

In addition, the existing studies that investigate the relation between education and non-cognitive skills provide mixed results. Some argue that education affects non-cognitive skills. Others claim that the impacts are significant for younger students, but not for the adolescents and adults. Additionally, it has been discussed that positive impacts of schooling on non-cognitive skills disappear after controlling for the previous levels of non-cognitive skills.

In this research, we apply a shift in Social Security students aid policy that took place in 1982 to study the impacts of an exogenous variation in years of schooling on non-cognitive skills. The reason for this approach is that the endogeneity of education is a well-known issue in studying the impacts of education on wages or intelligence. Applying an exogenous variation in years of schooling helps to overcome this issue. Hence, we apply a source of variation in schooling that is exogenous to unobservable variables that might influence non-cognitive abilities. We apply a shift in aid policy that affects the school attendance of some students, but not all of them to control for the endogeneity of education.

The study conducted by Heckman et al. (2013) is perhaps the most relevant study to ours. In their research, Heckman and his co-authors study the outcomes of the Perry Preschool Project (1962 to 1967) a randomized trial with a control group designed to provide preschool education to three to four years-old children living in poverty. The project had long-term effects on male income, adult crime rate, and test scores. However, it did not have any significant and lasting impact on IQ. Heckman et al. (2013) argue that the positive effects of the program are due to improvements in the non-cognitive abilities of the children. Additionally, Dee and West (2011) study the non-cognitive returns to class size and find that the reduction in the eighth grade class sizes in Germany is associated with an improvement in non-cognitive skills. Nevertheless, there has been doubt about the impacts of college education on non-cognitive skills, even though the impacts could be strong in younger ages. For example, Bachman et al. (1978) find that college education has little or no impact on self-esteem. Other researchers, such as Kautz et al. (2014), however, argue that non-cognitive skills are malleable in adolescent years and early adulthood and families and years of schooling influence them[1].

Psychologists have been studying personality traits[2] for a long time when compared with economists. When two influential studies by Heckman and co-authors were published, economists began paying attention to the non-cognitive literature. Nevertheless, the term "non-cognitive" has remained as controversial since then. The first of the two studies by Heckman and Rubinstein (2001) suggest that non-cognitive skills contribute to differences in the earnings of the two groups of GED recipients and high school graduates with similar degrees. They argue that the reason behind the difference in the income of the mentioned groups is that the GED recipients have a shortfall of non-cognitive skills as compared to the high school graduates. Heckman et al. (2006) applied the National Longitudinal Survey of Youth (NLSY) 1979 to study the importance of non-cognitive skills. They apply the two measurements of non-cognitive skills, including Rotter Locus of Control Scale and Rosenberg Self-Esteem Scale, as available in NLSY1979 dataset, and they suggest that non-cognitive abilities are equally as important as cognitive abilities in determining a variety of social and economic outcomes. Heckman et al. (2006) offer two paths through which non-cognitive abilities, including locus of control and self-esteem, could improve earnings. One path is direct effects on productivity in the labor-market and the other is through the indirect effects of non-cognitive abilities on schooling and work experience.

There has been debate in the literature regarding the importance of self-esteem in predicting life outcomes. In research that was conducted by Boden et al. (2007, 2008), the impacts of self-esteem on life outcomes became insignificant after they controlled for 23 covariates in their specifications. However, including that other researchers (Orth and Robins 2014) have criticized many control variables. Another study by Baumeister et al. (2003) argues that self-esteem is not a major predictor of educational and job performance. However, after that, Baumeister and co-authors published their research, many studies found self-esteem as a significant predictor of major life outcomes, as well as educational and job success (Trzesniewski et al. 2006; Orth et al. 2009; Orth et al. 2012; Kuster et al. 2013; Marshall et al. 2014). Additionally, in a review of the literature, Diener (1984) concludes that self-esteem is the most accurate predictor of life satisfaction in the United States, being stronger than physical health, age, education, income, and marital status. In addition, Orth and Robins (2014) argue that self-esteem predicts success and well-being in health outcomes, job market, and relationships. They maintain "given the increasing evidence that self-esteem has important real-world consequences, the topic of self-esteem development is of considerable societal significance" (Orth and Robins 2014, p. 381).

---

[1]  Kautz et al. (2014), particularly argue that non-cognitive skills are more malleable than cognitive skills in adolescent years.
[2]  Personality traits have been, to a high extent, considered heritable. Twin studies show that 40 to 60 percent of variation in non-cognitive skills and personality traits is due to genetic causes (Bouchard and Loehlin 2001). Nevertheless, they are under influence of environment, family background, and years of schooling as well.

The previous studies have discussed different ways that years of schooling can affect self-esteem[3]. For example, social comparison at school, which refers to comparing own work with those of the peers, might positively or negatively affect the self-esteem of the students. It is argued that social comparison serves several goals, such as self-presentation, mastery, self-assessment, and self-enhancement (Taylor and Lobel 1989; Butler 1989; Ruble and Frey 1991; Pomerantz et al. 1995). Social comparison boosts the student's self-esteem if they realize that their work is more complete than that of their classmates. However, if the students come to believe that they are less capable than their classmates then social comparison lowers their self-esteem (Pomerantz et al. 1995). Even, the damage to self-esteem of the students could be so serious that they might avoid it altogether. Brickman and Bulman (1977) argue that, despite the fact that social comparison might have a potential for self-enhancement, but it might be avoided because it can damage the self-esteem of own or others. For example, a student who looks at her classmate's work for self-assessment might be disappointed to know that she has accomplished far fewer homework assignments than her classmate. Additionally, a student who shows her finished work to her peers for the purpose of self-enhancement and realizes that they have not completed any of the assignment questions may feel embarrassed by her own superiority (Brickman and Bulman 1977).

Besides social comparison, school performance has been discussed to affect self-esteem. For example, Skaalvik and Hagtvet (1990) find that good performance at school one year leads to a higher self-esteem the year after. Additionally, Rosenberg et al. (1989) find a causality relationship from students' grades to self-esteem. They suggest that having a good performance at school make the students to think of themselves as capable at doing schoolwork, which can boost their self-esteem. Several studies find a correlation between school performance and self-esteem, but they often do not find any causality from self-esteem to school performance, even though they find causality from performance to self-esteem (Rosenberg et al. 1989; Skaalvik and Hagtvet 1990; Bowles 1999; Baumeister et al. 2003). Other studies, such as the one conducted by Rubin et al. (1977), suggest that the correlation between academic performance and self-esteem is based on important underlying factors, such as background and ability.

Another way that the impacts of years of schooling on self-esteem could be explained is through what that psychologists refer to as contingency of self-worth. Crocker and Wolfe (2001) define contingency of self-worth as "a domain or category of outcomes on which a person has staked his or her self-esteem, so that person's view of his or her value or worth depends on perceived successes or failures or adherence to self-standards in that domain" (Crocker and Wolfe 2001, p. 594). They argue that families, institutions, and social groups have implicit or explicit criteria for what is valuable and worthwhile in people. For example, universities value academic achievements. Over time, individuals internalize these criteria and they evaluate their own worth according to them (Crocker and Wolfe 2001). Success in achieving goals that are relevant to one's contingencies of self-worth may boost her self-esteem. For example, students might believe being smart is what makes them worthwhile. They try to get high grades, because it shows that they are smart. Subsequently, getting high grades improves their self-esteem. Additionally, when people believe a higher academic degree is what makes them worthwhile, receiving a higher academic degree might boost their self-esteem. However, people might have motivations that have little or nothing to do with contingency of self-worth. For example, a student might try to get high grades, or someone might try to receive a high academic degree only to facilitate landing a high paying job or to apply for a professional program (Crocker and Wolfe 2001; Crocker et al. 2002).

Locus of control, which is the other non-cognitive skill measurement in this paper, has also been positively associated with labor market outcomes (Heineck and Anger 2010; Cobb-Clark 2015; Caliendo et al. 2015). As an example, Heineck and Anger (2010) find that the external[4] locus of control

---

[3]　Self-esteem improves from adolescence to about age 50 to 60 years-old when it peaks. Then, it begins to decline at an accelerating rate until old age. Even though self-esteem is immutable, it is also relatively stable. In addition, someone with a high self-esteem might go through a stage of low self-esteem after a decade and vice versa (Orth and Robins 2014).

[4]　The meanings of internal and external locus of control are provided in Data and Identification Strategy section.

is associated with a decrease in wages. Caliendo et al. (2015) also argue that individuals with a more internal locus of control scale search harder for jobs when compared to the ones with a more external locus of control scale. Additionally, the individuals with internal locus of control scale have a stronger belief that investments in job search would pay off later. Additionally, individuals with external locus of control scale have lower reservation wages[5] (Caliendo et al. 2015).

It has been discussed in the literature that one of the most influential interventions on locus of control is achieving a higher social status (Wolfle and Robertshaw 1982). For example, Harvey (1971) maintains that upward mobility is associated with a change in locus of control toward internality. That means the individuals that experience an upward shift in their social status are likely to experience a change in their locus of control scale from external to internal levels. Particularly, education can create a path toward a higher social status and has the potential to affect locus of control toward a more internal level. However, it is worth noting that studies, such as the one by Wolfle and Robertshaw (1982), argue that the causal relationship between social status mobility and locus of control is a two-way street. Wolfle and Robertshaw (1982) emphasize that it is not surprising that new experiences, environments, and social conditions can lead to shifts in personality characteristics, because those who experience a shift in their status might have a high degree of internal control in the first place. Therefore, part of the correlation between upward social status mobility and personality characteristics can be attributed to the same personality characteristics that affect social mobility. In a review of the literature, Pascarella and Terenzini (1991) provide ample evidence that the locus of control increases during college education years (i.e., it becomes more internalized). However, Wolfle and Robertshaw (1982) suspect that the college students not only acquire higher education, but they are already internally oriented. Wolfle and Robertshaw (1982) argue that the strongest explanatory variable of the locus of control among high school students is ability such that those with higher abilities have more internalized locus of control. However, Wolfle and List (2004) results show that college education affects the locus of control of the students and shift it from external toward internal levels, even after controlling for ability, background, and previous levels of locus of control.

Non-cognitive skills[6] are not limited to the Rosenberg Self-Esteem Scale[7] and Rotter Locus of Control Scale[8]. They also include other skills. In this paper, we apply the Rotter Locus of Control Scale and Rosenberg Self-Esteem Scale for the exact same reason that Heckman and his co-authors use them. As they point out, "we choose these measures because of their availability in the NLSY79" (Heckman et al. 2006, p. 429). Ideally, using a wider range of psychological measurements is preferable. Nevertheless, the Rotter Locus of Control Scale and Rosenberg Self-Esteem Scale are two essential measurements of non-cognitive skills with significant educational and professional outcomes. Heckman et al. (2006) and other researchers, such as Waddell (2006) and Deke and Haimson (2006), argue that the Rotter Locus of Control Scale and Rosenberg Self-Esteem Scale are associated with long term labor market outcomes.

Numerous studies point out the importance of non-cognitive skills on labor market outcomes (e.g., Gintis 1971; Heckman and Rubinstein 2001; Bowles et al. 2001; Kuhn and Weinberger 2005; Heckman et al. 2006; Waddell 2006; Fortin 2008; Deke and Haimson 2006; Heineck and Anger 2010; Dee and West 2011; Drago 2011; de Araujo and Lagos 2013; de Araujo and Lagos 2013; Cobb-Clark 2015). A reasonable clue that shows the importance of non-cognitive skills in the job market comes from the

---

5    Reservation wage is defined as the lowest wage that an employee accepts for a particular job.
6    Note that psychologists and economists have used a variety of terminologies to describe non-cognitive skills. Other terms such as non-cognitive abilities, character skills, soft skills, socio-emotional skills, and personality traits have been used to refer to the same range of abilities or skills (Heckman and Kautz 2013; Garcia 2016). A well-known and widely accepted classification of personality traits (non-cognitive skills) is the "Big Five", which includes agreeableness, extraversion, neuroticism, conscientiousness, and openness to experience.
7    Rosenberg Self-Esteem Scale measures an individual's subjective belief about her/his own worth. More details on how this scale is measured is provided in Appendix A.
8    Rotter Locus of Control Scale measure the extend which individuals believe have control over their lives. More details on how this scale is measured is provided in Appendix B.

behavior of the employers. In a survey of 400 employers in the United States (U.S.), Casner-Lotto and Barrington (2006) find that the top four skills that employers look for in applicants are oral communication, teamwork, work ethic and professionalism, and problem solving and critical thinking. Writing abilities, mathematics knowledge, and other cognitively demanding skills ranked lower than the mentioned soft skills. Self-esteem, which is one of the non-cognitive measurements that we use in this paper, has been associated with teamwork and higher levels of perseverance, two top soft skills that employers seek in job candidates (Murnane et al. 2001). Another finding of Murnane et al. (2001) is that there is a positive association between self-esteem and earnings 10 years later. Additionally, Gintis (1971) argues that, if non-cognitive abilities are omitted in return to schooling specifications, then the estimated impacts of education on earnings would be biased due to the impacts of non-cognitive abilities on earnings.

Besides what is argued above, this study can help with an old debate in the literature: human capital versus signaling views. Positive impacts of education on non-cognitive skills is in favor of human capital view rather than signaling view. However, insignificant impacts of education on non-cognitive skills support the signaling view. According to the signaling view by Spence (1978), a school degree reveals the ability of a prospective employee to a potential employer by sending her a signal. Based on this view, which Michael Spence and other scholars have developed, credentials reveal the unobserved ability of the employees to the employers, which aids the employers to distinguish low and high ability workers from each other. After the employee enters the job market and starts working, the role of ability in determining the wages increases, because ability is more observable to the employer. Over time, the informational role of credentials in determining wages decreases because the employers can observe the employee's abilities. According to this view, people with higher levels of productivity choose higher levels of education to signal their ability to potential employers. Arcidiacono et al. (2010) apply the Armed Forces Qualification Test (AFQT) as a measure of ability and find that the ability of high school graduates is revealed gradually and over time to the job market, but the ability of college graduates is almost perfectly observed in the job market. The finding of this study suggest that a college degree might serve a signaling purpose. According to the human capital view that Gary Becker and other scholars developed, investment in education enhances productivity and, therefore, income. Becker argues that "Education, training, and health are the most important investments in human capital (Becker 1992, p. 1)". The supporters of human capital view argue that credentials do not serve employees merely due to their informational value in sending signals to the employers, but people improve their human capital by attending school (Becker 1964; Ben-Porath 1967).

As mentioned before, in this paper, we apply a shift in aid policy that took place in 1982 to study the impacts of an exogenous variation in years of schooling on non-cognitive abilities, including self-esteem and locus of control. Our results suggest that the impact of schooling on the Rosenberg Self-Esteem Scale and the Rotter Locus of Control Scale is such that it improves self-esteem and increases the sense of control over life (internal locus of control) when the previous levels of non-cognitive skills are not controlled for, but when those variables are included as the control variable then the results are hardly statistically significant.

The rest of this paper is organized, as follows. Section 2 explains the data and identification strategy. Section 3 outlines the empirical design of the paper. Section 4 provides the results and Section 5 presents the conclusion of the study.

## 2. Data and Identification Strategy

### 2.1. Data

NLSY79 is a nationally representative sample of 12,686 young American men and women aged between 14 and 22 years when they were first surveyed in 1979. After 1979, the data have been collected

in most of the years until 2016. Table 1 represents the number and percentage of each gender, and ethnic group is represented in NLSY79.

**Table 1.** Characteristics of NLSY79 dataset.

|  | **Number** | **Percentage** |
|---|---|---|
| Total | 12,686 | 100 |
| Hispanic | 2002 | 15.78 |
| Blacks | 3174 | 25.02 |
| Non-Hispanic & Black | 7510 | 59.19 |
| Males | 6403 | 50.47 |
| Females | 6283 | 49.53 |

We use the Rosenberg Self-Esteem Scale, which was designed by Rosenberg (1965), as one of the measurements of non-cognitive ability. NLSY79 provides the Rosenberg Self-Esteem Scale in the 1980, 1987, and 2006 interviews. In this scale, the respondents are asked to express their opinions regarding 10 statements of self-approval and disapproval. The respondents are asked to pick one of the options among strongly agree, agree, disagree, or strongly disagree. An example of a Rosenberg Self-Esteem Scale statement is "I am a person of worth". A higher score indicates higher self-esteem. Please see Appendix A for more details regarding the Rosenberg Self-Esteem Scale, including the questions that are asked from the participants, the number of participants, and the scores.

The Rotter Locus of Control Scale designed by Rotter (1966) is the other variable that we use as a measurement of non-cognitive skills. The NLSY79 provides Rotter Locus of Control Scale in 1979 and 2014. This variable measures the extent that individuals believe that they have control over their lives, which can be explained as internal control versus external control. Internal control is about the extent by which individuals believe that they have control over their lives through self-motivation or self-determination, while external control is about the extent that they believe the environment (that is, chance, fate, luck, etc.) controls them. Higher scores indicate that the individual is more external. However, we reverse the sign of locus of control scores in this paper, so the results are consistent with those of the self-esteem coefficients.

Note that we standardize self-esteem and locus of control scales, such that their means equal to zero and their standard deviations set to one. This enables us to interpret the impacts of education on the mentioned scales in percentage points.

*2.2. Identification Strategy*

Studying the impacts of education on abilities is a challenge, because an improvement in abilities could also improve education. In addition, unobserved variables that might affect both education and abilities contribute to the endogeneity of education. Therefore, the estimated impacts of education on ability could be biased due to the endogeneity of education. Nevertheless, exogenous variation in years of schooling helps to overcome the problem of endogeneity of education. We need a source of variation in schooling that is exogenous to unobservable variables that might influence non-cognitive abilities to overcome this issue. In this paper, we apply a shift in aid policy that affects the school attendance of some students but not all of them to study the impacts of schooling on non-cognitive abilities. The elimination of a Social Security student benefit program in 1982 is the policy that we apply.

From 1965 to 1982, the Social Security Administration provided aid for the 18 to 22 years old children of Social Security beneficiaries, including the children of deceased, disabled, and retired parents. Prior to 1965, aid was provided to the children of Social Security beneficiaries only up to age 18. From 1965 to 1982, the Social Security Administration covered students up to age 22 if they were full time college students. According to the data from National Center for Education Statistics, the program covered 700,000 college students at its peak. From 1977 to 1982, the number of covered students dropped moderately to 600,000, due to a national drop in college enrolment rates in the U.S.

(Dynarski 2003). In 1981, Congress voted to eliminate the Social Security student benefit program. Students who were not enrolled in college by May 1982 were no longer eligible to receive aid. Moreover, payments to the ones who were enrolled in college were significantly reduced.

We apply eligibility for aid due to death of the parent in this paper since parental retirement and disability could be endogenous due to availability of student benefits. Following Dynarski (2003), we focus on the students with deceased fathers since 90 percent of the students were eligible for the program from that cause. Note that the focus of the paper is estimating the impacts of aid eligibility on education and non-cognitive abilities, rather than estimating the impacts of aid receipt on the mentioned variables. This approach enables us to unbiasedly predict the impact of eliminating the aid policy, because the policy makers control the offer of the aid but not its take up.

Table 2 shows the mean and standard error of the education level of the students with deceased and not deceased father before and after the policy change. Schooling by age 23 is the number of years of schooling that a person has completed by age 23. The variable complete college by age 23 is a binary variable that sets one if an individual completed any years of college by age 23. Otherwise, it is equal to zero. Complete college by age 28 equals to one if a person completed any years of college before age 28. For each variable listed in Table 2, the number of observations is provided underneath the mean and standard error. Standard errors are in parentheses. The mean of all the variables in the table show that children of deceased fathers are better off in terms of education before the change in policy in 1982. For example, their years of schooling by age 23 that has been 13 and larger than that of other students has decreased to 12.7, while years of schooling of the regular students has increased. The last column of Table 2 shows the difference-in-differences impact of eligibility for Social Security student benefits on educational outcomes. The positive means in the last column show the positive impact of the student eligibility for the Social Security student aid program on education outcomes. Standard errors in this column are clustered by household, due to the presence of siblings in some families. Table 2 provides some suggestive evidence for the difference-in-differences findings and the results in this table will be tested in the next sections of the paper.

**Table 2.** NLSY Summary Statistics.

| | High School Seniors 1979–1981 | | High School Seniors 1982–1983 | | |
| --- | --- | --- | --- | --- | --- |
| | Father Not Deceased | Father Deceased | Father Not Deceased | Father Deceased | Difference-in-Differences |
| schooling by age 23 | 12.839 (0.050) | 13 (0.090) | 12.852 (0.095) | 12.702 (0.175) | 0.309 (0.215) |
| Number of obs. | 699 | 247 | 210 | 74 | 1230 |
| complete college by age 23 | 0.372 (0.018) | 0.425 (0.031) | 0.394 (0.033) | 0.358 (0.054) | 0.089 (0.071) |
| Number of obs. | 716 | 256 | 218 | 81 | 1271 |
| complete college by age 28 | 0.461 (0.019) | 0.505 (0.031) | 0.552 (0.034) | 0.419 (0.055) | 0.177 (0.074) |
| Number of obs. | 722 | 257 | 219 | 81 | 1279 |

Note: Schooling by age 23 is the number of years of schooling that an individual has completed by age 23. Complete college by age 23 is a binary variable that equals to one if an individual has completed any years of college by age 23. Otherwise, it is equal to zero. Complete college by age 28 equals to one if a person completed any years of college by age 28. Otherwise, it is equal to zero.

## 3. Empirical Design

We apply the difference-in-differences approach to study the impacts of eligibility for the Social Security benefits on educational outcomes. To do so, we estimate the following standard difference-in-differences model:

$$Schooling_i = \beta_1 + \beta_2(father\ deceased_i \times before_i) + \beta_3 father\ deceased_i + \beta_4 before_i + \beta_5 X_i + \epsilon_i \quad (1)$$

where $Schooling_i$ stands for educational outcomes for individual $i$. In the empirical results section of the paper, when we show the results, we clarify what exactly $Schooling_i$ stands for in each estimated regression. *Father deceased* is a binary variable that is equal to one for those who are potentially eligible for Social Security student benefits, due to death of their fathers. Also $before_i$, is a binary variable that shows that individual $i$ graduated from high school before the elimination of the program. $before_i$ equals to one if individual $i$ is a high school senior in 1979, 1980, or 1981. The NLSY79 dataset that we use starts in 1979. Therefore, we do not have data prior to 1979. $before_i$ is zero if individual $i$ is a high school senior in 1982 or 1983 (i.e., after elimination of the program). The reason that high school seniors after 1983 are not included in our regressions is that they age out of the 18–22 years old eligibility for the program. $X_i$ represents a vector of family background control variables and the birth year and region fixed effects.

We apply instrumental variables specifications to address this issue since the literature discusses that schooling could be endogenous due to unobserved innate ability. In a regular least square specification that aims to estimate the impacts of education on labor market outcomes, innate ability is in the error term and it might be correlated with the dependent variable.

We are interested in understanding the impacts of years of schooling on non-cognitive skills (i.e., Rosenberg Self-Esteem Scale and Rotter Locus of Control Scale). This relationship can be specified, as follows:

$$Y_i = \delta_1 + \delta_2 Schooling_i + \delta_3 father\ deceased_i + \delta_4 before_i + \delta_5 X_i + \epsilon_i \tag{2}$$

where $Y_i$ is the outcome variable for individual $i$. The outcome variable in our regressions is either the Rosenberg Self-Esteem Scale or Rotter Locus of Control Scale. The rest of the variables are introduced before.

We apply the interaction between *father deceased* and *before* as an instrument in the following model, which serves as the first stage of our two-stage least-square estimations specification for Equation (2):

$$Schooling_i = \theta_1 + \theta_2 (father\ deceased_i \times before_i) + \theta_3 father\ deceased_i + \theta_4 before_i + \theta_5 X_i + \omega_i \tag{3}$$

where all the variables are introduced in Equations (1) and (2). The interaction of *father deceased* and *before* is an exogenous variable that affects years of education, but it does not directly affect noncognitive abilities.

## 4. Empirical Results

In this section, Table 3 provides the estimates of the difference-in-differences model (i.e., Equation (1)) and Tables 4–7 present the estimates of the least-square and the two-stage least-square models (i.e., Equations (2) and (3)). The goal in Table 3 is to estimate the impacts of aid eligibility on schooling. This helps us to realize whether the elimination of the Social Security student benefits causes any variations in schooling outcomes. In Tables 4 and 5, the least-square methodology is applied to estimate the impacts of schooling on non-cognitive abilities. As discussed before, the results from least-square estimations could be biased due to the possible endogeneity of schooling. Therefore, in Tables 6 and 7, the two-stage least-square estimations that apply instrumental variables to control for the endogeneity of schooling are provided. The two-stage least-square regressions estimate the impacts of variation in years of schooling on non-cognitive abilities.

**Table 3.** Impact of eligibility for Social Security student benefits on education.

| | Schooling by Age 23 | | Complete College by Age 23 | | Complete College by Age 28 | |
|---|---|---|---|---|---|---|
| father deceased × before | 0.309 | 0.491 * | 0.089 | 0.238 ** | 0.177 ** | 0.293 *** |
| | (0.215) | (0.259) | (0.071) | (0.085) | (0.074) | (0.088) |
| father deceased | −0.149 | −0.348 | −0.036 | −0.180 ** | −0.132 ** | −0.257 *** |
| | (0.199) | (0.235) | (0.063) | (0.075) | (0.064) | (0.078) |
| before | −0.012 | 0.311 | −0.021 | −0.503 ** | −0.091 ** | −0.118 |
| | (0.107) | (0.884) | (0.037) | (0.244) | (0.038) | (0.295) |
| family income | | Y | | Y | | Y |
| black | | Y | | Y | | Y |
| Hispanic | | Y | | Y | | Y |
| father attended college | | Y | | Y | | Y |
| mother attended college | | Y | | Y | | Y |
| single parent family | | Y | | Y | | Y |
| family size | | Y | | Y | | Y |
| age88 | | Y | | Y | | Y |
| female | | Y | | Y | | Y |
| age^2 | | Y | | Y | | Y |
| married | | Y | | Y | | Y |
| nosmsa | | Y | | Y | | Y |
| centrsmsa | | Y | | Y | | Y |
| Zasvab | | Y | | Y | | Y |
| Year of birth fixed effect | | Y | | Y | | Y |
| Region fixed effect | | Y | | Y | | Y |
| All covariates × before | | Y | | Y | | Y |
| Number of obs. | 1230 | 875 | 1279 | 884 | 1279 | 884 |
| R-squared | 0.003 | 0.189 | 0.046 | 0.171 | 0.005 | 0.177 |

Note: Schooling by age 23 is the number of years of schooling that an individual has completed by age 23. Complete college by age 23 is a binary variable that equals to one if an individual has completed any years of college by age 23. Otherwise, it is equal to zero. Complete college by age 28 equals to one if a person completed any years of college by age 28. Otherwise, it is equal to zero. *: Significant at 10%; **: Significant at 5%; ***: Significant at 1%.

**Table 4.** Impact of education on non-cognitive skills—least-square estimations. Independent variable of interest: schooling by age 23.

| Dependent Variable | (1) | N. of obs. | R2 | (2) | N. of obs. | R2 | (3) | N. of obs. | R2 | (4) | N. of obs. | R2 | (5) | N. of obs. | R2 | (6) | N. of obs. | R2 |
|---|---|---|---|---|---|---|---|---|---|---|---|---|---|---|---|---|---|---|
| Z. Rosenberg 1987 | 0.135 *** (0.019) | 1173 | 0.045 | 0.127 *** (0.023) | 869 | 0.069 | 0.127 *** (0.023) | 869 | 0.071 | 0.136 *** (0.023) | 869 | 0.082 | 0.127 *** (0.023) | 858 | 0.090 | 0.140 *** (0.023) | 858 | 0.140 |
| Z. Rosenberg 2006 | 0.064 *** (0.022) | 883 | 0.011 | 0.050 * (0.027) | 640 | 0.038 | 0.050 * (0.027) | 640 | 0.039 | 0.054 * (0.028) | 640 | 0.042 | 0.050 * (0.027) | 631 | 0.071 | 0.048 (0.029) | 631 | 0.118 |
| Z. Rotter 2014 | 0.095 *** (0.024) | 760 | 0.021 | 0.072 ** (0.031) | 557 | 0.041 | 0.073 ** (0.031) | 557 | 0.041 | 0.079 ** (0.031) | 557 | 0.050 | 0.076 * (0.031) | 549 | 0.060 | 0.075 ** (0.031) | 549 | 0.146 |

Note: The upper numbers in columns (1) to (6) show the coefficient of schooling by age 23. Column (1) include no control variables. Column (2) includes father deceased, before, family income, black, Hispanic, father attended college, mother attended college, single parent family, family size, age88, and female. Column (3) includes all the covariates in column (2) and age powered by 2. Column (4) includes all the covariates in column (3) and married, nosmsa, centrsmsa, Zasvab. Column (5) includes all the covariates in column (4) and year of birth and region fixed effect. Column (6) includes all of the covariates in column (5) and the interaction of all covariates and *before*. Family income is the income of the family that individual $i$ has grown up there. Black is a binary variable that set to one if individual $i$ is black and it is zero otherwise. Hispanic is one if individual $i$ is Hispanic and it is zero otherwise. Father and mother attended college are binary variables that show parents of individual $i$ have attended college. Single parent is a binary variable that equals to one if individual $i$ has grown up in a single parent household and it equals to zero otherwise. Family size refers to actual number of the family members. Female is a binary variable that sets to one if individual $i$ is a female and it is zero otherwise. Age squared is age powered by two. Married, is a binary variable and shows whether a person is married or single. Nonsmsa sets to one if individual $i$ does not live in a city, and it is zero otherwise. Centrsmsa equals to one if individual $i$ lives in the central city. Zasvab is the standardized ASVAB score of individual $i$, Region of birth refers to one of the four regions in the United States: Northeast, the Midwest, the South, and the West. All covariates × before is all covariates multiplied by *before*. The numbers in parentheses represent standard errors. N. of obs. represents number of observations. R2 stands for R squared. All regressions are clustered at the household level. *: Significant at 10%; **: Significant at 5%; ***: Significant at 1%.

**Table 5.** Impact of education on non-cognitive skills–least-square estimations. Independent variable of interest: complete college by age 23.

| Dependent Variable | (1) | N. of obs. | R2 | (2) | N. of obs. | R2 | (3) | N. of obs. | R2 | (4) | N. of obs. | R2 | (5) | N. of obs. | R2 | (6) | N. of obs. | R2 |
|---|---|---|---|---|---|---|---|---|---|---|---|---|---|---|---|---|---|---|
| Z. Rosenberg 1987 | 0.381 *** (0.054) | 1188 | 0.045 | 0.362 *** (0.064) | 876 | 0.070 | 0.362 *** (0.064) | 876 | 0.070 | 0.362 *** (0.064) | 876 | 0.072 | 0.386 *** (0.065) | 876 | 0.083 | 0.366 *** (0.065) | 865 | 0.092 |
| Z. Rosenberg 2006 | 0.169 *** (0.058) | 898 | 0.010 | 0.160 ** (0.072) | 646 | 0.038 | 0.163 ** (0.072) | 646 | 0.040 | 0.170 ** (0.073) | 646 | 0.042 | 0.154 ** (0.073) | 637 | 0.072 | 0.143 * (0.078) | 637 | 0.113 |
| Z. Rotter 2014 | 0.306 *** (0.066) | 774 | 0.027 | 0.324 *** (0.083) | 562 | 0.056 | 0.326 *** (0.083) | 562 | 0.057 | 0.339 *** (0.083) | 562 | 0.065 | 0.329 *** (0.084) | 554 | 0.075 | 0.315 *** (0.086) | 554 | 0.156 |

Note: The upper numbers in columns (1) to (6) show the coefficient of complete college by age 23. Column (1) include no control variables. Column (2) includes father deceased, before, family income, black, Hispanic, father attended college, mother attended college, single parent family, family size, age88, and female. Column (3) includes all the covariates in column (2) and age powered by 2. Column (4) includes all the covariates in column (3) and married, nosmsa, centrsmsa, Zasvab. Column (5) includes all the covariates in column (4) and year of birth and region fixed effect. Column (6) includes all the covariates in column (5) and the interaction of all covariates and *before*. Family income is the income of the family that individual *i* has grown up there. Black is a binary variable that set to one if individual *i* is black and it is zero otherwise. Hispanic is one if individual *i* is Hispanic and it is zero otherwise. Father and mother attended college are binary variables that show parents of individual *i* have attended college. Single parent is a binary variable that equals to one if individual *i* has grown up in a single parent household and it equals to zero otherwise. Family size refers to actual number of the family members. Female is a binary variable that sets to one if individual *i* is a female and it is zero otherwise. Age squared is age powered by two. Married, is a binary variable and shows whether a person is married or single. Nonsmsa sets to one if individual *i* does not live in a city, and it is zero otherwise. Centrsmsa equals to one if individual *i* lives in the central city. Zasvab is the standardized ASVAB score of individual *i*, Region of birth refers to one of the four regions in the United States: Northeast, the Midwest, the South, and the Wes All covariates × before t. is all covariates multiplied by *before*. The numbers in parentheses represent standard errors. N. of obs. represents number of observations. R2 stands for R squared. All regressions are clustered at the household level. *: Significant at 10%; **: Significant at 5%; ***: Significant at 1%.

**Table 6.** Impact of education on non-cognitive skills—two-stage least-square estimations. Instrumented variable: schooling by age 23.

| Independent Variable | Z. Rosenberg 1987 | Z. Rosenberg 2006 | Z. Rotter 2014 |
|---|---|---|---|
| Coefficient of schooling by age 23 | 0.269 ** (0.105) | 0.245 ** (0.100) | 0.274 * (0.158) |
| Number of obs. | 896 | 667 | 576 |
| Test of weak instruments: first stage F | 11.933 | 11.297 | 14.960 |
| Previous non-cognitive measure as a control variable | NO | NO | NO |
| Coefficient of schooling by age 23 | 0.115 (0.114) | 0.189 * (0.110) | 0.262 (0.165) |
| Number of obs. | 895 | 666 | 573 |
| Test of weak instruments: first stage F | 9.435 | 9.350 | 14.143 |
| Previous non-cognitive measure as a control variable | Z. Rosenberg 1980 | Z. Rosenberg 1980 | Z. Rotter 1979 |

Instrumented: schooling by age 23. Instrument: interaction of *father deceased* and *before*. Test of weak instrument Ho: Instruments are weak. Control variables: family income, age, and standardized ASVAB score. All regressions are clustered at the household level. The numbers in parentheses represent standard errors. *: Significant at 10%; **: Significant at 5%; ***: Significant at 1%.

**Table 7.** Impact of education on non-cognitive skills—two-stage least-square estimations. Instrumented variable: complete any years of college by age 23.

| Independent Variable | Z. Rosenberg 1987 | Z. Rosenberg 2006 | Z. Rotter 2014 |
|---|---|---|---|
| Coefficient of complete college by age 23 | 0.918 ** (0.378) | 0.907 ** (0.360) | 0.928 * (0.489) |
| Number of obs. | 905 | 676 | 581 |
| Test of weak instruments: first stage F | 9.761 | 9.308 | 8.808 |
| Previous non-cognitive measure as a control variable | NO | NO | NO |
| Coefficient of complete college by age 23 | 0.351 (0.426) | 0.700 * (0.401) | 0.850 * (0.471) |
| Number of obs. | 904 | 675 | 562 |
| Test of weak instruments: first stage F | 7.023 | 7.142 | 11.746 |
| Previous non-cognitive measure as a control variable | Z. Rosenberg 1980 | Z. Rosenberg 1980 | Z. Rotter 1979 |

Instrumented: complete any years of college by age 23. Instrument: interaction of *father deceased* and *before*. Test of weak instrument Ho: Instruments are weak. Control variables: family income, age, and standardized ASVAB score. All regressions are clustered at the household level. The numbers in parentheses represent standard errors. *: Significant at 10%; **: Significant at 5%; ***: Significant at 1%.

In Table 3, the results for the main difference-in-differences model are estimated with either no control variables or a full set of control variables. Note that Appendix C provides estimations of the difference-in-differences model with four different subsets of control variables. In Table 3, age squared is age powered by two and shows the sensitivity of the results to the functional form of the regression taken by age. Additionally, Nonsmsa is a binary variable that sets to one if individual *i* does not live in a city and it is zero otherwise. Centrsmsa is a binary variable that equals one if individual *i* lives in the central city. Zasvab is the standardized ASVAB score of individual *i*, The ASVAB includes 10 tests that measure skill and knowledge in general science, arithmetic reasoning, paragraph comprehension, word knowledge, coding speed, numerical operations, mathematics knowledge, auto and shop information, electronics information, and mechanical comprehension. A standardized ASVAB score has been added to our covariates to control for knowledge and cognitive abilities of the participants. The region of birth refers to one of the four regions in the United States: Northeast, the Midwest, the South, and the West. 'All covariates × before' is the interaction of all covariates with *before*. This controls for heterogeneity across eligibility status and time. For example, this interaction term absorbs the changes in economy that might affect family income and, therefore, school attendance of the youth.

Table 3 represents the impact of eligibility for student benefits on years of schooling by age 23, complete college by age 23, and complete college by age 28. Note that schooling is not a binary variable, but it is the actual years of schooling. However, complete college by age 23 and 28 are the binary variables. The results do not show the significant impacts of eligibility for student benefits on years of schooling by age 23 and completing any years of college by age 23 when no control variable is included in the regressions. However, in most cases, we see positive and significant impacts when the control

variables are included. These results could be interpreted as positive impacts of the student benefits program on education outcomes. However, these findings should be interpreted with caution because, as mentioned above, the impacts are not significant when the control variables are not included. Another reason is that NLSY79 dataset does not include data prior to 1979. This might also affect the findings that are presented in Table 3. Please see Appendix C for more details.

Family income is the income of the family that individual $i$ has grown up there. Black is a binary variable that set to one if individual $i$ is black and it is zero otherwise. Hispanic is one if individual $i$ is Hispanic and it is zero otherwise. Father and mother attended college are the binary variables that show parents of individual $i$ have attended college. Single parent is a binary variable that equals to one if individual $i$ has grown up in a single parent household and it equals zero otherwise. Family size refers to actual number of the family members. Female is a binary variable that sets to one if individual $i$ is a female and it is zero otherwise. Age squared is age powered by two. Married, is a binary variable and it shows whether a person is married or single. Nonsmsa sets to one if individual $i$ does not live in a city, and it is zero otherwise. Centrsmsa equals to one if individual $i$ lives in the central city. Zasvab is the standardized ASVAB score of individual $i$, The region of birth refers to one of the four regions in the United States: Northeast, the Midwest, the South, and the West. All covariates × before is all covariates multiplied by *before*.

The upper numbers in each column show the coefficient of the independent variable provided in the first column (i.e., column in the left) of the table. The numbers in parentheses represent the standard errors. Y means that the associated covariate is included in the estimation of the equation. All of th regressions are clustered at the household level. *: Significant at 10%; **: Significant at 5%; ***: Significant at 1%.

As mentioned before, the goal of this paper is to understand the impacts of schooling on non-cognitive abilities, rather than the impacts of aid on schooling. Dynarski (2003) studied the impacts of variation in aid on years of schooling and college attendance. Here, we not only replicated her results and found similar impacts of aid on education, but we provided more details on the impacts of aid eligibility on schooling. Our findings to this point give us confidence that we have estimated the impacts of eligibility for Social Security student benefits on schooling correctly. We apply this variation in our two-stage least-square estimations specifications to estimate the impacts of the years of education on non-cognitive abilities. However, we have presented the results of least-square estimations before discussing the two-stage least-square estimations results.

Table 4 shows the results of the least-square estimations of Equation (2). The upper numbers in each cell under columns (1) to (6) show the coefficient of schooling by age 23. The numbers in parentheses represent the standard errors. Column (1) is estimated without any control variables. The covariates in column (2) include family Income, black, Hispanic, father attended college, mother attended college, single parent, family size, age, and female. The covariates in column (3) include all of those in column (2) and age powered by 2. In column (4), all of the covariates of column (3) and four new covariates, including married, nosmsa, centrsmsa, and standardized ASVAB test scores, are added to the regressions. In column (5), all of the covariates in column (4) and year of birth and region of birth fixed effects are added. In column (6), all covariates in column (5) and the interaction of all of covariates with *before* are included.

The first row in the table shows the impacts of schooling by age 23 on the standardized Rosenberg Self-Esteem Scale that is measured in 1987. Note that, in 1987, the youngest participants in NLSY 1979 dataset are 23 years old. The results show that the impact of years of schooling by age 23 on the standardized Rosenberg Self-Esteem Scale that was measured in 1987 is positive and relatively stable in columns (1) to (6). The impacts vary from 12.7 to 14.0 percentage points.

The second row shows the impact of years of schooling completed by age 23 on the standardized Rosenberg Self-Esteem Scale measured in 2006. In 2006, the youngest participant in the NLSY dataset is 42 years old. Hence, other than some rare exceptions, everybody should be out of school by 2006. Here, the coefficients are smaller when compared to those in the first row and we still see some

positive impacts of years of schooling on the Rosenberg Self-Esteem Scale measured in 2006 when the participants ages range from 42 to 49. Nevertheless, note that the coefficients are only statistically significant at 10 percent level in columns (1) to (5) and they are not statistically significant in column (6).

The last row in Table 4 represents the impacts of schooling by age 23 on the Standardized Rotter Locus of Control Scale that was measured in 2014. As can be seen, the coefficients are positive, which suggest that more years of schooling is associated with an increase in the sense of control over life (positive impacts on internal locus of control and negative impacts on external locus of control).

Table 5 represents the impacts of complete any years of college by age 23 on the standardized Rosenberg Self-Esteem Scale that was measured in 1987 and 2006 and standardized Rotter Locus of Control Scale measured in 2014. The results show the positive impacts of complete any years of college by age 23 on non-cognitive abilities. Note that results from least square regressions could be biased due to the endogeneity of education.

Tables 6 and 7 show the two-stage least-square estimations of Equation (3). We have used the interaction of *father deceased* and *before* as an instrument. This instrument has good explanatory power in the first stage. Note that using the interactions between *father deceased* and *after* as an instrument produces exact same results with the same sign as that of the interaction of *father deceased* and *before*. In Table 6 the instrumented variable is schooling by age 23 and in Table 7 the instrumented variable is completed any years of college by age 23. Tables A13–A18 in Appendix D provide the results of the two-stage least-square estimations with different subsets of control variables.

The dependent variable in Table 6 is the standardized Rosenberg Self-Esteem Scale that was measured in 1987. The results are presented in two rows. In the first row, the previous levels of non-cognitive abilities are not controlled for, but they are controlled for in the second row.

Perhaps the most important aspect of the results in Tables 6 and 7 is that education has a significant impact on non-cognitive skills when we do not control for the previous levels of non-cognitive skill, but we hardly see any significant impact when the prior levels of non-cognitive skills are controlled for. In Tables 6 and 7, when a previous level of self-esteem is not included as a control variable, we can see a significant and positive impact of education on self-esteem and when a previous measure of self-esteem is included as a control variable, the impacts are still positive, but they either are not statistically significant or they are only significant at the 10 percent level. The impact of education on locus of control is positive and statistically significant at the 10 percent level when a previous level of locus of control is not controlled for, but when it is controlled for then the impact is not statistically significant.

## 5. Conclusions

Understanding the true impacts of college education on non-cognitive skills remains a highly crucial question in the literature, as the previous studies have not paid enough attention to this issue, despite the fact that they have highlighted the importance of non-cognitive skills. Multiple studies argue that non-cognitive skills are highly important in daily life and they have crucial life and professional outcomes. For example, Diener (1984) finds self-esteem to be the strongest predictor of life satisfaction in the United States, stronger than physical health, age, education, income, and marital status. Nevertheless, the previous studies do not offer a clear picture of the impacts of education on non-cognitive skills. Heckman et al. (2013) finds positive impacts of education on non-cognitive abilities of three to four years-old children and Dee and West (2011) finds out that reduction in the eighth grade class sizes in Germany improves the non-cognitive skills. Does college education have positive impacts on non-cognitive skills, as Heckman et al. (2013) and Dee and West (2011) find for children in younger ages? Kautz et al. (2014), argue that non-cognitive skills are malleable in adolescent years and they are influenced by schooling. However, Bachman et al. (1978) maintain that college education has little or no impact on self-esteem. Bachman et al. (1978) find that educational attainment does not affect self-esteem when previous levels of self-esteem are controlled for, and Wolfle and Robertshaw (1982) reach an identical finding regarding locus of control. Wolfle and Robertshaw (1982) results show that education beyond high school does not significantly affect locus of control. Nevertheless, Pascarella

and Terenzini (1991) maintain that the locus of control improves during college education years and becomes more internalized, but Wolfle and Robertshaw (1982) suspect that the college students are already internally oriented. They argue that the strongest explanatory variable of locus of control among high school students is ability, such that those with higher abilities have more internalized locus of control. However, Wolfle and List (2004) results show that college education affects the locus of control of the students and shift it from external toward internal levels, even after controlling for ability, background, and previous levels of locus of control.

We apply NLSY79 that contains data from over 12,000 young Americans to study the impacts of college education in the United States on non-cognitive abilities. A variation in the years of schooling that is exogenous to unobservable variables that might influence non-cognitive abilities is essential for overcoming the problem of endogeneity of schooling, since schooling is an endogenous variable. In this paper, we apply a shift in aid policy that affects school attendance of a portion of students, but not all of them to capture exogenous variations in years of schooling. The policy that we apply is the elimination of Social Security student benefits program in 1982, which, from 1965 to 1982, provided aid for the children of Social Security beneficiaries aged 18 to 22.

Note that the focus of the paper is to estimate the impacts of aid eligibility on educational outcomes and non-cognitive abilities rather than the impacts of aid receipt on mentioned variables. Additionally, we focus on eligibility for aid due to the death of father. 90 percent of the students were eligible for the program, because of the death of their fathers. In our two-stage least-square estimations, we have used the interaction of *father deceased* and *before* as an instrument. The interaction of *father deceased* and *before* is an exogenous variable that affects years of education, but does not directly affect non-cognitive abilities. *Father deceased* is a binary variable that indicates whether the father is deceased, and *before* is another binary variable that shows that a student has been a high school senior in the years 1979, 1980, and 1981versus years 1982 and 1983.

Our difference-in-differences results suggest a positive impact of eligibility for student benefits on years of schooling and completing any years of education at college, but these findings should be interpreted with caution due to limitations of NLSY79 dataset, as mentioned before.

The results of the two-stage least-square estimates show that education has a significant impact on non-cognitive skills when the prior levels of non-cognitive skills are not controlled for, but, when the prior levels of non-cognitive skills are controlled for, then we hardly see any significant impacts of education on non-cognitive skills. The results show that the impact of education on self-esteem is significant and positive when a previous level of self-esteem is not included as a control variable, and the impacts are still positive when a previous level of self-esteem is included as a control variable, but in most cases they are not statistically significant.

The results from the two-stage least-square estimations show parallel findings for the impacts of education on locus of control. In most cases, when the previous levels of locus of control are included, the statistically significance level of the coefficients decreases. The coefficients stay positive in all estimations, which suggest that the locus of control shifts from external to internal levels during college years.

Note that the results of this paper do not entirely deny the positive impacts of formal college education on non-cognitive abilities. We also find positive impacts, but when the prior levels of non-cognitive skills are control for, we rarely see any significant impact. Hence, our results are in line with the findings of Wolfle and Robertshaw (1982) and Pascarella and Terenzini (1991), who found similar results decades ago. As mentioned before, several scholars report positive impacts of education on non-cognitive skills, which is a different finding from that of Wolfle and Robertshaw (1982) and Pascarella and Terenzini (1991). Our results do not confirm any causality relationship from college education to non-cognitive skills.

Note that we do not deny the potential positive impacts of the Social Security student aid program on the life and professional outcomes of the beneficiary students. We find positive impacts of the program on education attainment and the program possibly had other positive impacts that could be

investigated in the future studies. Additionally, note that this paper does not provide any cost-benefit analysis of the Social Security student aid program. In addition, the sample that was used in this paper, which includes the children of Social Security beneficiaries at high school and college age, might not represent all American students at all levels of education. However, we believe that these results are replicable for samples of students that are more representative of the population of American college students.

**Funding:** This research received no external funding.

**Acknowledgments:** I thank Teresa Molina, Sang-Hyop Lee, Timothy Halliday, Nori Tarui, Jennifer Darrah, Michael Roberts, John Lynham, Inessa Love, and Ekaterina Sherstyuk, for valuable inputs and comments. I especially appreciate the guide and support of Andrew Mason. Also, I am thankful of two anonymous reviewers as their comments improved the quality of the paper drastically. I bear sole responsibility for the content of this research and all errors remain mine.

**Conflicts of Interest:** The author declare no conflict of interest.

## Appendix A. Rosenberg Self-Esteem Scale in NLSY1979

In this appendix, information regarding Rosenberg Self-Esteem Scale in NLSY1979 database has been provided. The following sentence from NLSY1979 and Table A1 show the question that has been asked in the 1987 and 2006 interviews:

"Now I'm going to read a list of opinions people have about themselves. After I read each statement, please tell me how much you strongly agree, agree, disagree or strongly disagree with these opinions".

**Table A1.** Rosenberg Self-Esteem Scale questions in NLSY1979 database.

| Number | |
|--------|---|
| 1 | I feel that I'm a person of worth, at least on equal basis with others |
| 2 | I feel that I have a number of good qualities |
| 3 | All in all, I am inclined to feel that I am a failure |
| 4 | I am able to do things as well as most other people |
| 5 | I feel I do not have much to be proud of |
| 6 | I take a positive attitude toward myself |
| 7 | On the whole, I am satisfied with myself |
| 8 | I wish I could have more respect for myself |
| 9 | I certainly feel useless at times |
| 10 | At times I think I am no good at all |

Tables A2–A5 represent data on the respondents of the Rosenberg Self-Esteem Scale questionnaire in NLSY1979. Table A2 shows the number of respondents picked each option from the Rosenberg Self-Esteem Scale questions in a 1987 interview.

**Table A2.** Data on Rosenberg Self-Esteem Scale respondents 1987.

| Number of the Question | Strongly Agree | Agree | Disagree | Strongly Disagree | Total |
|:----------------------:|:--------------:|:-----:|:--------:|:-----------------:|:------:|
| 1 | 5780 | 4501 | 155 | 22 | 10,458 |
| 2 | 5689 | 4674 | 82 | 20 | 10,465 |
| 3 | 40 | 233 | 4736 | 5450 | 10,459 |
| 4 | 4807 | 5451 | 150 | 40 | 10,448 |
| 5 | 127 | 478 | 4696 | 5140 | 10,441 |
| 6 | 4346 | 5671 | 380 | 46 | 10,443 |
| 7 | 3313 | 6347 | 729 | 66 | 10,455 |
| 8 | 299 | 2102 | 5226 | 2812 | 10,439 |
| 9 | 163 | 2340 | 5236 | 2714 | 10,453 |
| 10 | 82 | 873 | 5175 | 4328 | 10,458 |

Table A3 shows the number of respondents in the 1987 interview, along with their associated Rosenberg Self-Esteem Scale scores.

**Table A3.** Respondents in 1987 interview along with their associated Rosenberg Self-Esteem Scale scores.

| Number of Respondents | Score |
|:---:|:---:|
| 0 | 0 |
| 2 | 1 TO 4 |
| 4 | 5 TO 9 |
| 104 | 10 TO 14 |
| 1796 | 15 TO 19 |
| 4209 | 20 TO 24 |
| 3226 | 25 TO 29 |
| 999 | 30 TO 34 |
| - - - - - - - | |
| 10,340 | |
| Min: 3          Max: 30 | Mean: 23.37 |

Table A4 shows the number of respondents who picked each option from Rosenberg Self-Esteem Scale questions in 2006 interview.

**Table A4.** Data on Rosenberg Self-Esteem Scale respondents 2006.

| Number of the Question | Strongly Agree | Agree | Disagree | Strongly Disagree | Total |
|:---:|:---:|:---:|:---:|:---:|:---:|
| 1 | 4195 | 3225 | 159 | 31 | 7610 |
| 2 | 4271 | 3285 | 54 | 15 | 7625 |
| 3 | 44 | 241 | 3538 | 3747 | 7570 |
| 4 | 3329 | 3865 | 365 | 56 | 7615 |
| 5 | 119 | 364 | 3498 | 3573 | 7554 |
| 6 | 3256 | 4032 | 291 | 37 | 7616 |
| 7 | 2483 | 4408 | 649 | 70 | 7610 |
| 8 | 219 | 1295 | 3628 | 2386 | 7528 |
| 9 | 110 | 1261 | 3685 | 2494 | 7550 |
| 10 | 57 | 485 | 3636 | 3357 | 7535 |

Table A5 shows the number of respondents in the 2006 interview, along with their associated Rosenberg Self-Esteem Scale scores.

**Table A5.** Respondents in 2006 interview along with their associated Rosenberg Self-Esteem Scale scores.

| Number of Respondents | Score |
|:---:|:---:|
| 0 | 0 |
| 0 | 1 TO 4 |
| 12 | 5 TO 9 |
| 131 | 10 TO 14 |
| 1195 | 15 TO 19 |
| 2834 | 20 TO 24 |
| 2277 | 25 TO 29 |
| 920 | 30 TO 34 |
| - - - - - - - | |
| 7369 | |
| Min: 5          Max: 30 | Mean: 23.48 |

## Appendix B. Rotter Locus of Control in NLSY1979

In this appendix, information regarding Rotter Locus of Control Scale in NLSY1979 database has been provided. The following sentence from NLSY1979 and Tables A6 and A7 show the questions that were asked in the 2014 interview:

"I am going to read you four pairs of statements about outlooks on life. For each pair, please select one statement which is closer to your opinion. In addition, tell me whether the statement you select is much closer to your opinion or slightly closer. Try to consider each pair of statements separately when making your choice; do not be influenced by your previous choices".

**Table A6.** Rotter Locus of Control Scale questions in NLSY1979 database.

| Item | Question |
|------|----------|
| 1 | Pair 1, statement A |
| 2 | Pair 1, statement B |
| 3 | Pair 2, statement A |
| 4 | Pair 2, statement B |
| 5 | Pair 3, statement A |
| 6 | Pair 3, statement B |
| 7 | Pair 4, statement A |
| 8 | Pair 4, statement B |

Table A7 shows the number of respondents who picked each option related to Rotter Locus of Control Scale questions in the 2014 interview. Note that pair 1. B is asked after pair 1. A is asked. Additionally, pair 2. B, pair 3. B, and pair 4. B are asked after pair 2. A, pair 3. A, pair 4. A are asked, respectively.

Pair 1. B to pair 4. B ask a same question which is as follow:

"Is this statement much closer or slightly closer to your opinion?"

**Table A7.** Data on Rotter Locus of Control respondents 2014.

| | Question | | Number of Responses | | |
|---|---|---|---|---|---|
| | **Statement A** | **Statement B** | **Statement A** | **Statement B** | **Total** |
| Pair 1 | What happens to me is my own doing. | Sometimes I feel that I don't have enough control over the direction my life is taking. | 5784 | 1186 | 6970 |
| Pair 2 | When I make plans, I am almost certain that I can make them work. | It is not always wise to plan too far ahead, because many things turn out to be a matter of good or bad fortune anyhow. | 5004 | 2001 | 7005 |
| Pair 3 | In my case, getting what I want has little or nothing to do with luck. | Many times, we might just as well decide what to do by flipping a coin. | 6070 | 835 | 6905 |
| Pair 4 | Many times, I feel that I have little influence over the things that happen to me. | It is impossible for me to believe that chance or luck plays an important role in my life. | 3184 | 3613 | 6797 |

Table A8 shows the number of respondents picked each option from Rotter Locus of Control Scale. Pair B questions in 2014 interview.

**Table A8.** Data on Locus of Control respondents 2014: pair B.

| Question | Number of Responses | | |
|----------|---------------------|---|---|
| | **1. Much Closer** | **2. Slightly Closer** | **Total** |
| Pair 1. B. | 4863 | 2080 | 6943 |
| Pair 2. B. | 4794 | 2188 | 6982 |
| Pair 3. B. | 4058 | 2826 | 6884 |
| Pair 4. B. | 3184 | 3600 | 6784 |

Table A9 shows the number of respondents in the 2014 interview along with their associated Rotter Locus of Control Scale scores.

**Table A9.** Rotter Locus of Control Scale respondents in 2014 interview along with their associated scores.

| Score | Number of Respondents |
|---|---|
| 1 TO 4 | 738 |
| 5 TO 9 | 4522 |
| 10 TO 14 | 1316 |
| 15 TO 19 | 39 |
| Total | 6615 |
| Min: 4 Max: 16 Mean: 7.56 | |

## Appendix C

The findings in Table 3 are from the estimates of the difference-in-differences model. They show the impact of eligibility for Social Security student benefits on education. Several variables are included in the list of the control variables in Table 3. Tables A10–A12 in this appendix provide the results of the estimates of the same model while applying different subsets of control variables that are not provided in Table 3.

Each of Tables A10–A12 contain four columns. The control variables in column (1) of each table include family income, black, Hispanic, father has attended college, mother has attended college, single parent, family size, age in 1988, and female. In column (2), age squared is added to the equation. This shows the sensitivity of the results to the functional form of the regression taken by age. In column (3), we have added four new control variables, including married, which is a binary variable that shows whether a person is married or single. Also, Nonsmsa is included in column (3). Nonsmsa sets to one if individual $i$ does not live in a city, and it is zero otherwise. Another covariate in column (3) is Centrsmsa, which equals one if individual $i$ lives in the central city. Zasvab is the standardized ASVAB score of individual $i$, which has been added to NLSY79 dataset in 1979. The ASVAB includes 10 tests that measure skill and knowledge in general science, arithmetic reasoning, paragraph comprehension, word knowledge, coding speed, numerical operations, mathematics knowledge, auto and shop information, electronics information, and mechanical comprehension. In the estimations that are provided in columns (3) and (4), a standardized ASVAB score has been added to our covariates to control for the skills, knowledge, and cognitive abilities of the participants. In column (4), the year of birth fixed effects and region fixed effects are added. The region of birth refers to one of the four regions in the United States: Northeast, the Midwest, the South, and the West.

**Table A10.** Impact of eligibility for Social Security student benefits on years of schooling by age 23.

| Schooling by Age 23 | (1) | (2) | (3) | (4) |
|---|---|---|---|---|
| father deceased × before | 0.577 ** | 0.572 ** | 0.561 *** | 0.611 ** |
| | (0.230) | (0.231) | (0.234) | (0.240) |
| father deceased | −0.392 * | −0.387 * | −0.422 ** | 0.467 ** |
| | (0.206) | (0.207) | (0.211) | (0.216) |
| before | 0.584 *** | 0.545 *** | 0.528 *** | 0.657 *** |
| | (0.156) | (0.163) | (0.163) | (0.161) |
| family income | Y | Y | Y | Y |
| black | Y | Y | Y | Y |
| Hispanic | Y | Y | Y | Y |
| father attended college | Y | Y | Y | Y |
| mother attended college | Y | Y | Y | Y |
| single parent family | Y | Y | Y | Y |
| family size | Y | Y | Y | Y |
| age88 | Y | Y | Y | Y |
| female | Y | Y | Y | Y |
| age^2 | | Y | Y | Y |
| married | | | Y | Y |
| nosmsa | | | Y | Y |
| centrsmsa | | | Y | Y |
| Zasvab | | | Y | Y |
| Year of birth fixed effect | | | | Y |
| Region fixed effect | | | | Y |
| All covariates × before | | | | |
| Number of obs. | 886 | 886 | 886 | 875 |
| R-squared | 0.118 | 0.119 | 0.147 | 0.157 |

Note: The dependent variable is years of schooling by age 23. This is the actual number of years of schooling if the respondents in NLSY79 dataset has completed by age 23. The upper numbers in columns (1) to (4) show the coefficient of the independent variable provided in the first column (i.e., column in the left) of the table. The numbers in parentheses represent standard errors. Y means that the associated covariate is included in the estimation of the equation. All regressions are clustered at the household level. *: Significant at 10%; **: Significant at 5%; ***: Significant at 1%.

**Table A11.** Impact of eligibility for Social Security student benefits on complete any years of college by age 23.

| Complete College by Age 23 | (1) | (2) | (3) | (4) |
|---|---|---|---|---|
| father deceased × before | 0.224 *** | 0.224 *** | 0.224 *** | 0.251 *** |
| | (0.079) | (0.079) | (0.080) | (0.081) |
| father deceased | −0.153 * | −0.154 ** | −0.167 ** | −0.192 ** |
| | (0.069) | (0.069) | (0.070) | (0.071) |
| before | 0.135 *** | 0.141 ** | 0.131 ** | 0.140 *** |
| | (0.050) | (0.058) | (0.058) | (0.058) |
| family income | Y | Y | Y | Y |
| black | Y | Y | Y | Y |
| Hispanic | Y | Y | Y | Y |
| father attended college | Y | Y | Y | Y |
| mother attended college | Y | Y | Y | Y |
| single parent family | Y | Y | Y | Y |
| family size | Y | Y | Y | Y |
| age88 | Y | Y | Y | Y |
| female | Y | Y | Y | Y |
| age^2 | | Y | Y | Y |
| married | | | Y | Y |
| nosmsa | | | Y | Y |
| centrsmsa | | | Y | Y |
| Zasvab | | | Y | Y |
| Year of birth fixed effect | | | | Y |
| Region fixed effect | | | | Y |
| All covariates × before | | | | |
| Number of obs. | 895 | 895 | 895 | 884 |
| R-squared | 0.093 | 0.093 | 0.121 | 0.129 |

Note: The dependent variable is complete any years of college by age 23. This is a binary variable that sets to one if individual *i* has completed any years of college by age 23. It equals to zero otherwise. The upper numbers in columns (1) to (4) show the coefficient of the independent variable provided in the first column (i.e., column in the left) of the table. The numbers in parentheses represent standard errors. Y means that the associated covariate is included in the estimation of the equation. All regressions are clustered at the household level. *: Significant at 10%; **: Significant at 5%; ***: Significant at 1%.

**Table A12.** Impact of eligibility for Social Security student benefits on complete any years of college by age 28.

| Complete College by Age 28 | (1) | (2) | (3) | (4) |
|---|---|---|---|---|
| father deceased × before | 0.316 *** | 0.318 *** | 0.302 *** | 0.325 *** |
| | (0.082) | (0.082) | (0.082) | (0.083) |
| father deceased | −0.259 * | −0.261 ** | −0.260 ** | −0.283 ** |
| | (0.072) | (0.072) | (0.072) | (0.073) |
| before | 0.121 *** | 0.141 ** | 0.138 *** | 0.144 *** |
| | (0.050) | (0.059) | (0.059) | (0.061) |
| family income | Y | Y | Y | Y |
| black | Y | Y | Y | Y |
| Hispanic | Y | Y | Y | Y |
| father attended college | Y | Y | Y | Y |
| mother attended college | Y | Y | Y | Y |
| single parent family | Y | Y | Y | Y |
| family size | Y | Y | Y | Y |
| age88 | Y | Y | Y | Y |
| female | Y | Y | Y | Y |
| age^2 | | Y | Y | Y |
| married | | | Y | Y |
| nosmsa | | | Y | Y |
| centrsmsa | | | Y | Y |
| Zasvab | | | Y | Y |
| Year of birth fixed effect | | | | Y |
| Region fixed effect | | | | Y |
| All covariates × before | | | | |
| Number of obs. | 895 | 895 | 895 | 884 |
| R-squared | 0.114 | 0.115 | 0.139 | 0.150 |

Note: The dependent variable is complete any years of college by age 28. This is a binary variable that sets to one if individual *i* has completed any years of college by age 28. It equals to zero otherwise. The upper numbers in columns (1) to (4) show the coefficient of the independent variable provided in the first column (i.e., column in the left) of the table. The numbers in parentheses represent standard errors. Y means that the associated covariate is included in the estimation of the equation. All regressions are clustered at the household level. *: Significant at 10%; **: Significant at 5%; ***: Significant at 1%.

## Appendix D

Tables 6 and 7 show the two-stage least-square estimations of Equation (3). In this appendix, the results of the two-stage least-square estimations are provided while applying different subsets of control variables that are not provided in Tables 6 and 7. In Tables A13–A15, the instrumented variable is schooling by age 23, but in Tables A16–A18 the instrumented variable is complete any years of college by age 23. The interaction of *father deceased* and *before* is applied as our instrument in all regressions.

**Table A13.** Impact of education on Rosenberg Self-Esteem 1987—two-stage least-square estimations. Instrumented variable: schooling by age 23.

| Independent Variable | (1) | (2) | (3) | Previous Non-Cognitive Variable is Control Variable |
|---|---|---|---|---|
| Z. Rosenberg 1987 | 0.287 ** | 0.304 ** | 0.260 ** | NO |
| | (0.113) | (0.117) | (0.118) | |
| Number of obs. | 869 | 869 | 858 | |
| Test of weak instruments: first stage F | 10.784 | 10.037 | 10.919 | |
| Z. Rosenberg 1987 | 0.130 | 0.148 | 0.119 | Z. Rosenberg 1980 |
| | (0.121) | (0.125) | (0.125) | |
| Number of obs. | 868 | 868 | 857 | |
| Test of weak instruments: first stage F | 8.731 | 8.046 | 9.034 | |

Instrumented: schooling by age 23. Instrument: interaction of *father deceased* and *before*. Test of weak instrument Ho: Instruments are weak. Control variables in column (1) include family income, black, Hispanic, father attended college, mother attended college, single parent family, family size, age, standardized ASVAB score, and female. Column (2) includes all the covariates in column (1) as well as married, nosmsa, and centrsmsa. Column (3) includes all of the covariates in column (2) and year and region fixed effects. All regressions are clustered at the household level. The numbers in parentheses represent standard errors. *: Significant at 10%; **: Significant at 5%; ***: Significant at 1%.

**Table A14.** Impact of education on Rosenberg Self-Esteem 2006–two-stage least-square estimations. Instrumented variable: schooling by age 23.

| Independent Variable | (1) | (2) | (3) | Previous Non-Cognitive Variable is Control Variable |
|---|---|---|---|---|
| Z. Rosenberg 2006 | 0.224 ** | 0.217 ** | 0.131 | NO |
| | (0.108) | (0.108) | (0.116) | |
| Number of obs. | 640 | 640 | 631 | |
| Test of weak instruments: first stage F | 9.901 | 9.787 | 11.382 | |
| Z. Rosenberg 2006 | 0.161 | 0.157 | 0.067 | Z. Rosenberg 1980 |
| | (0.117) | (0.117) | (0.128) | |
| Number of obs. | 639 | 639 | 630 | |
| Test of weak instruments: first stage F | 8.372 | 8.295 | 9.609 | |

Instrumented: schooling by age 23. Instrument: interaction of *father deceased* and *before*. Test of weak instrument Ho: Instruments are weak. Control variables in column (1) include family income, black, Hispanic, father attended college, mother attended college, single parent family, family size, age, standardized ASVAB score, and female. Column (2) includes all of the covariates in column (1) as well as married, nosmsa, and centrsmsa. Column (3) includes all the covariates in column (2) and year and region fixed effects. All regressions are clustered at the household level. The numbers in parentheses represent standard errors. *: Significant at 10%; **: Significant at 5%; ***: Significant at 1%.

**Table A15.** Impact of education on Rotter Locus of Control 2014—two-stage least-square estimations. Instrumented variable: schooling by age 23.

| Independent Variable | (1) | (2) | (3) | Previous Non-Cognitive Variable is Control Variable |
|---|---|---|---|---|
| Z. Rotter 2014 | 0.259 * (0.155) | 0.242 (0.151) | 0.224 (0.137) | NO |
| Number of obs. | 557 | 557 | 549 | |
| Test of weak instruments: first stage F | 15.246 | 15.136 | 15.301 | |
| Z. Rotter 2014 | 0.247 (0.158) | 0.232 (0.156) | 0.220 (0.141) | Z. Rotter 1979 |
| Number of obs. | 554 | 554 | 546 | |
| Test of weak instruments: first stage F | 15.520 | 15.342 | 15.087 | |

Instrumented: schooling by age 23. Instrument: interaction of *father deceased* and *before*. Test of weak instrument Ho: Instruments are weak. Control variables in column (1) include family income, black, Hispanic, father attended college, mother attended college, single parent family, family size, age, standardized ASVAB score, and female. Column (2) includes all of the covariates in column (1) as well as married, nosmsa, and centrsmsa. Column (3) includes all the covariates in column (2) and year and region fixed effects. All regressions are clustered at the household level. The numbers in parentheses represent standard errors. *: Significant at 10%; **: Significant at 5%; ***: Significant at 1%.

**Table A16.** Impact of education on Rosenberg Self-Esteem 1987—two-stage least-square estimations. Instrumented variable: complete any years of college by age 23.

| Independent Variable | (1) | (2) | (3) | Previous Non-Cognitive Variable is Control Variable |
|---|---|---|---|---|
| Z. Rosenberg 1987 | 0.905 ** (0.388) | 0.972 ** (0.403) | 0.820 ** (0.405) | NO |
| Number of obs. | 876 | 876 | 865 | |
| Test of weak instruments: first stage F | 9.859 | 9.364 | 7.943 | |
| Z. Rosenberg 1987 | 0.354 (0.425) | 0.418 (0.446) | 0.330 (0.444) | Z. Rosenberg 1980 |
| Number of obs. | 875 | 875 | 864 | |
| Test of weak instruments: first stage F | 7.548 | 7.083 | 6.306 | |

Instrumented: complete any years of college by age 23. Instrument: interaction of *father deceased* and *before*. Test of weak instrument Ho: Instruments are weak. Control variables in column (1) include family income, black, Hispanic, father attended college, mother attended college, single parent family, family size, age, standardized ASVAB score, and female. Column (2) includes all the covariates in column (1) as well as married, nosmsa, and centrsmsa. Column (3) includes all the covariates in column (2) and year and region fixed effects. All regressions are clustered at the household level. The numbers in parentheses represent standard errors. *: Significant at 10%; **: Significant at 5%; ***: Significant at 1%.

**Table A17.** Impact of education on Rosenberg Self-Esteem 2006—two-stage least-square estimations. Instrumented variable: complete any years of college by age 23.

| Independent Variable | (1) | (2) | (3) | Previous Non-Cognitive Variable is Control Variable |
|---|---|---|---|---|
| Z. Rosenberg 2006 | 0.824 ** (0.382) | 0.800 ** (0.386) | 0.467 (0.356) | NO |
| Number of obs. | 646 | 646 | 637 | |
| Test of weak instruments: first stage F | 8.841 | 8.882 | 9.913 | |
| Z. Rosenberg 2006 | 0.588 (0.414) | 0.569 (0.419) | 0.252 (0.391) | Z. Rosenberg 1980 |
| Number of obs. | 645 | 645 | 636 | |
| Test of weak instruments: first stage F | 7.076 | 7.111 | 8.164 | |

Instrumented: complete any years of college by age 23. Instrument: interaction of *father deceased* and *before*. Test of weak instrument Ho: Instruments are weak. Control variables in column (1) include family income, black, Hispanic, father attended college, mother attended college, single parent family, family size, age, standardized ASVAB score, and female. Column (2) includes all of the covariates in column (1) as well as married, nosmsa, and centrsmsa. Column (3) includes all the covariates in column (2) and year and region fixed effects. All regressions are clustered at the household level. The numbers in parentheses represent standard errors. *: Significant at 10%; **: Significant at 5%; ***: Significant at 1%.

**Table A18.** Impact of education Rotter Locus of Control 2014—two-stage least-square estimations. Instrumented variable: complete any years of college by age 23.

| Independent Variable | (1) | (2) | (3) | Previous Non-Cognitive Variable is Control Variable |
|---|---|---|---|---|
| Z. Rotter 2014 | 0.850 * (0.471) | 0.794 * (0.465) | 0.674 (0.414) | NO |
| Number of obs. | 562 | 562 | 554 | |
| Test of weak instruments: first stage F | 11.746 | 12.372 | 12.765 | |
| Z. Rotter 2014 | 0.812 * (0.491) | 0.761 (0.489) | 0.660 (0.432) | Z. Rotter 1979 |
| Number of obs. | 559 | 559 | 551 | |
| Test of weak instruments: first stage F | 11.206 | 11.807 | 12.489 | |

Instrumented: complete any years of college by age 23. Instrument: interaction of *father deceased* and *before*. Test of weak instrument Ho: Instruments are weak. Control variables in column (1) include family income, black, Hispanic, father attended college, mother attended college, single parent family, family size, age, standardized ASVAB score, and female. Column (2) includes all the covariates in column (1) as well as married, nosmsa, and centrsmsa. Column (3) includes all the covariates in column (2) and year and region fixed effects. All regressions are clustered at the household level. The numbers in parentheses represent standard errors. *: Significant at 10%; **: Significant at 5%; ***: Significant at 1%.

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
