# Peer review of "Does Schooling Causally Impact Non-Cognitive Skills? Evidence from Elimination of Social Security Student Benefits"

_economies, doi:10.3390/economies8010005_

Round 1

Reviewer 1 Report

Summary

The paper uses the removal of financial aid for colleges as an instrument to show that years of schooling increases self-esteem and internal locus of control. The paper addresses an important research question and has a relatively convincing identification strategy. However, as described below, there are a number of issues in terms of the presentation.

Major comments

-“However, family and years of schooling also affect non-cognitive skills”. This statement should be corroborated, either through corresponding literature or through an explanation why years of schooling affect non-cognitive skills. Does Heckman (2013) analyze self-esteem and locus of control? What does he find? In contrast, for my taste, the argument that self-esteem and locus of control matters, could be shortened.

-The first stages in Tables 3 and 4 change substantially from model 1 to model 2. Hence, the instrument appears not to be orthogonal to these observables. This needs to be discussed as it raises concerns regarding the exclusion restriction of the instrument. Furthermore, the 2SLS estimates suggest that the OLS estimates underestimate the true effect. Hence, there is a negative selection in terms of self-esteem and locus of control into education?

-Tables 3 and 4 use four definitions of schooling (years by 23, years by 28, college completion by 23, college completion by 28). Table 5 uses two of those definitions (years by 23, years by 28). Table 6 uses one of those definitions (years by 23) and Table 7 appears to introduce a new definition (college enrolment by 23). It remains unclear to me, why these differences arise.

Minor comments

-Please mention in the data section that you standardize the main dependent variables. Does this imply that the coefficients are interpreted in terms of standard deviations? Please explain how this relates to your interpretation in terms of percentage points.

-Please provide notes for Table 2.

-Why does formula (2) exclude “deceased” and “before”? For Table 5, this may not matter, but it does for Tables 6 and 7, raising the question why formula (3) excludes these baseline effects as well?

-Does Table 4 refer to years of schooling and if so, why is the title different? Assuming, they are the same, I suggest to integrate Table 3 into Table 4 (or the other way round as I find Table 3 more clearly presented). In the latter case, please show coefficients of father deceased and before in Table 3 for all models.

-Observations differ across specifications, which would ideally be trimmed to the same data set at least for the same dependent and main independent variable, i.e. each row of your tables.

-Why are the number of observations so big in Table 5? I assume that you are using the full sample here but focus in the other tables on high school seniors between 1979 and 1983? I fail to comprehend the description of this on p.7.

-Why does Table 5 exclude models 1 and 6 from Tables 3 and 4? Beside of being interesting by themselves, this makes the paper harder to read.

-Please mention in the notes how errors are specified in Tables 6 and 7.

-Please mention the control variables X in Tables 6 and 7 that are suggested by formulas (2) and (3). Did you test different specifications similar as in Table 5?

-I do not understand the meaning of an overidentifying restriction in a model with a single instrument.

-The data contains the locus of control in 1979 and self-esteem in 1980. Could you not use these to control for the initial level of the variables, apply a fixed effect approach or to run a Placebo estimation?

-Spelling mistakes I spotted:

p.7 deceeased

p.9 Table 4 Similar

p.13 week instruments

Author Response

Dear reviewer,

We truly appreciate your contribution to this research. We found your comments very important and relevant and we have addressed every single one of them. We believe the quality of the paper has drastically improved in the light of your comments. Most importantly, the results suggest that non-cognitive skills improve during the college education years, but the causality relationship from college education to non-cognitive skills disappear when the previous levels of non-cognitive skills are controlled for. Considering the previous debate in the literature these results seem very interesting to us.

We upload a word document with detailed information on how we addressed your comments. 

Thanks again 

Reviewer 2 Report

The authors deserve credit for asking a relatively underexplored question regarding the impacts of years of education on noncognitive skills and using a unique instrumental variables strategy to identify an exogenous source of variation. The authors also deserve credit for clearly explaining their methods. It was easily to follow what they wished to do and understand why they needed to do it. However, I still have some serious doubts about their results and suggestions that need to be addressed before the paper is further considered for publication.

1) The introduction is quite detailed and the authors include much of the relevant literature. However, there are places where I think the authors are too verbose. The long direct quotes, in particular, can be removed and reduced to a summary statement. I suggest some edits to improve brevity because there are three issues that the authors should address to improve the introduction.

First, the authors take 4 pages before mentioning any detail about they are going to do in the paper. Mentioning the key question, method, and results of the paper closer to the beginning of the paper would improve readability.

Second, the introduction could benefit from a discussion of theory. Specifically, why should we expect years of education to influence self-esteem, locus of control, or other noncognitive skills? What is it about the educational process that would lead to changes in noncognitive skills.  It is not enough to simply cite Kautz et al (2014) and other claims that noncognitive skills are shaped by schools. Nor is it enough to simply identify educational programs like Perry Preschool and their impacts on noncognitive skills. With a large contingent of scholars who argue that many noncognitive skills are not malleable, theory for why further education improves noncognitive skills needs to be discussed in tandem with empirical results. Otherwise, the paper comes across as atheoretical empiricism or post-hoc rationalization.

Third, I would advise the authors to discuss the literature more summarily. There are some places where the authors get into the details of several studies and the manuscript begins to read like a long, mechanical list of prior studies. Rather than going into the details of each study, a simply summary statement of the main conclusions and their bearing on the authors’ present work would make the introduction more readable and offer space to theoretically reason why further education would affect noncognitive skills.

2) The empirical strategy is well explained but I have three major concerns with it.

One source of skepticism that I have stems from the first stage of their IV (Equation 3). The authors’ primary purpose is to isolate exogenous variation in years of education. Given that noncognitive skills are endogenous to years of education — a point the authors extensively emphasize, they should at least control for baseline measures of locus of control and self-esteem in the first stage. It seems like the X vector does not include those variables, and those variables are available: the authors on page 5 indicate that locus of control was measured in 1980 and locus of control in 1979, both of which are measured prior to the social security policy change.

The first stage also seems misspecified for a second reason. Unless I’m missing something, the authors should include dummies for a deceased father and pre-policy period as in Equation 1. Without these dummies, Ćź2 captures the difference in years of schooling between pre-policy individuals with a deceased father and everyone else. However, having a deceased father would seem to be a traumatic event that affects noncognitive skills and educational attainment. Hence, it seems that the authors need to parse out differences between pre-policy individuals with a deceased father, post-policy individuals with a deceased father, and other individuals across both policy periods without a deceased father. Alternatively, the authors could leave Equation 3 as is and restrict their sample to individuals across both time periods with a deceased father. It is not clear that they use this restricted sample given the sample sizes in Table 1 and Tables 6 and 7.

Thirdly, I’m skeptical that the change in college enrollment pre and post the Social Security policy shift is necessarily due to the policy shift. Ideally, a diff-in-diff would should secular trends for several years pre-policy change and then a jump. Given the data limitations of NLSY, this doesn’t seem to be possible. The authors also mention a Dynarski piece on page 6 that documents decreasing college enrollment in the years leading up to the policy shift. It’s possible that the enrollment difference pre- and post-policy are nothing more than these secular trends and have nothing to do with an exogenous shock from a policy change, which would undermine the crux of the identification strategy.

3)Moving onto results, the authors can be much more succinct by eliminating columns 2-5 in Tables 3 and 4. I also think they can eliminate results that use “schooling by age 28” as the key independent variable. Given that the results are robust to different specifications, simply presenting the parsimonious and kitchen-sink models (Columns 1 and 6) and using “schooling by age 23” seems sufficient.

Stylistically, I would also reverse code the locus of control variable so that the “positive” result has a positive sign and is consistent with the self-esteem coefficients.

5) Given some of the skepticism that I have, it may behoove the authors to tone down the causal language or at least acknowledge the limitations up front. For example, in the section on the identification strategy, the might acknowledge how their IV might fall short of isolating causal effects. Similarly, acknowledging limits to external validity seem appropriate here. By narrowly examining people with deceased fathers, it is difficult to generalize results to the broader population.

4) Minor typos: The authors reference “week” instruments when they should be “weak” instruments.

Author Response

Dear reviewer,

We truly appreciate your contribution to this research. We found your comments very important and relevant and we have addressed every single one of them. We believe the quality of the paper has drastically improved in the light of your comments. Most importantly, the results suggest that non-cognitive skills improve during the college education years, but the causality relationship from college education to non-cognitive skills disappear when the previous levels of non-cognitive skills are controlled for. Considering the previous debate in the literature these results seem very interesting to us.

We upload a word document that contains detailed information on how we have addressed your comments. 

Thanks again 

Round 2

Reviewer 1 Report

The authors have addressed my concerns.

Author Response

Thanks again for reviewing the paper. 

Reviewer 2 Report

I must applaud the authors for their diligence in addressing all the comments from my earlier review. I was originally skeptical whether the authors could sufficiently address all of my critiques while still maintaining the integrity of the original paper.  However, they seem to have addressed all of these critiques sufficiently. The paper is much improved and I am much more comfortable seeing this paper progress towards publication. But before then, I still have one general reaction for the authors to address.  My broad suggestion is to see the authors temper their tone and be more forthcoming about the ambiguities and issues that their analysis do not fully address.

In particular, the change in the results of the paper once controlling for prior levels of noncognitive abilities is quite fascinating and tempers the claim that schooling improves these noncognitive skills. It becomes less clear whether schooling improves human capital or simply serves as a signal of individual abilities. In the perennial debate about nature versus nurture, the results suggest that the truth lies somewhere in between where both are factors to explaining life outcomes. The authors hint at the ambiguity in discussing their results, and I would like to push the authors to highlight this much more.

On my reading of the revised version, I sense the authors still want to make a strong conclusion that education improves noncognitive skills. Given the ambiguities underlying the results, the authors should temper the entire paper and come across as more skeptical.  Emphasizing more detail about the signaling vs. human capital theory of education would go a long way to tempering the paper appropriately. Notably, this would also balance out the literature review, which is dominated by theoretical reasons for the human capital side of the debate. Emphasizing both sides of the debate and a more skeptical tone would greatly improve the paper.

Similarly, I would like to see the authors temper their claim that eligibility for Social Security benefits had a positive impact on college enrollment. Given the absence of pre-trends and the potential of confounding, secular trends that explain decreases in college enrollment before and after the policy, the authors should refrain from using causal language to link social security benefits and college enrollment. Besides, making this causal claim is not necessary for their primary claim about whether education affects noncognitive skills or not. Indeed, I raised a similar comment in the previous round of reviews and the authors do not seem to have addressed it.  I’ve pasted my comment from the earlier review here:

“Thirdly, I’m skeptical that the change in college enrollment pre and post the Social Security policy shift is necessarily due to the policy shift. Ideally, a diff-in-diff would should secular trends for several years pre-policy change and then a jump. Given the data limitations of NLSY, this doesn’t seem to be possible. The authors also mention a Dynarski piece on page 6 that documents decreasing college enrollment in the years leading up to the policy shift. It’s possible that the enrollment difference pre- and post-policy are nothing more than these secular trends and have nothing to do with an exogenous shock from a policy change...”

In conclusion, the authors tackle a question that has scholarly significance. Their analysis will hardly be the last word on the topic. Between (1) the sensitivity of the results to models with and without baseline measures of noncognitive skills and (2) lingering doubts as to how exogenous a Social Security policy shock really is, readers would not unreasonable to see this study as not providing compelling evidence about education affecting noncognitive skills. There is nothing wrong to publishing a paper with these limitations, but an appropriate tone in which claims are not overstated and comport to the empirical results are warranted as others continue to study the topic.

Author Response

Dear reviewer,

Thanks again for reading the paper and giving us valuable comments. We truly appreciate your contribution to this research. Again, we have done our best to address all of your comments.  

I must applaud the authors for their diligence in addressing all the comments from my earlier review. I was originally skeptical whether the authors could sufficiently address all of my critiques while still maintaining the integrity of the original paper.  However, they seem to have addressed all of these critiques sufficiently. The paper is much improved and I am much more comfortable seeing this paper progress towards publication. But before then, I still have one general reaction for the authors to address.  My broad suggestion is to see the authors temper their tone and be more forthcoming about the ambiguities and issues that their analysis do not fully address.

In particular, the change in the results of the paper once controlling for prior levels of noncognitive abilities is quite fascinating and tempers the claim that schooling improves these noncognitive skills. It becomes less clear whether schooling improves human capital or simply serves as a signal of individual abilities. In the perennial debate about nature versus nurture, the results suggest that the truth lies somewhere in between where both are factors to explaining life outcomes. The authors hint at the ambiguity in discussing their results, and I would like to push the authors to highlight this much more.

On my reading of the revised version, I sense the authors still want to make a strong conclusion that education improves noncognitive skills. Given the ambiguities underlying the results, the authors should temper the entire paper and come across as more skeptical.  Emphasizing more detail about the signaling vs. human capital theory of education would go a long way to tempering the paper appropriately. Notably, this would also balance out the literature review, which is dominated by theoretical reasons for the human capital side of the debate. Emphasizing both sides of the debate and a more skeptical tone would greatly improve the paper.

We have done our best to temper the entire paper. Starting from the abstract, we have changed the abstract from the following:

Abstract: In this paper, we apply the elimination of Social Security student benefits that took place in the United States in 1982 to study the impacts of education on non-cognitive skills measured by Rotter Locus of Control Scale and Rosenberg Self-Esteem Scale. We apply eligibility for aid due to death of father to avoid endogeneity in our analysis. The results confirm that the Social Security student aid program had positive impacts on educational outcomes. Also, our results suggest that non-cognitive skills improve during the college education years, but the causality relationship from college education to non-cognitive skills to a high extent disappear when the previous levels of non-cognitive skills are controlled for. Limited number of studies have investigated the impacts of education on non-cognitive skills, yet they offer mixed results. A few studies suggest that schooling does not significantly affect self-esteem or locus of control, yet other researchers report positive impacts of education on either self-esteem or locus of control. The elimination of Social Security student benefits enables us to investigate the causal impact of schooling on self-esteem and locus of control. Hence the results of this paper shed light on the impacts of schooling on non-cognitive skills.

To the following:

Limited number of studies have investigated the impacts of education on non-cognitive skills, yet they offer mixed results. A few studies suggest no impact, yet others report positive impacts of education on non-cognitive skills. In this paper, we apply the elimination of Social Security student benefits that took place in the United States in 1982 to study the impacts of education on non-cognitive skills measured by Rotter Locus of Control Scale and Rosenberg Self-Esteem Scale. We apply eligibility for aid due to death of father to avoid endogeneity in our analysis. Our results suggest that non-cognitive skills improve during the college education years, but the causality relationship from college education to non-cognitive skills to a high extent disappear when the prior levels of non-cognitive skills are controlled for.

Also, we added the following paragraph to emphasize more details about the signaling vs. human capital views:

Beside what argued above, yet another importance of this study is that it can help with an old debate in the literature: human capital versus signaling views. Positive impacts of education on non-cognitive skills is in favor of human capital view rather than signaling view. Yet, insignificant impacts of education on non-cognitive skills suggest that signaling view might be closer to reality. According to the signaling view by Spence (1978), a school degree reveals the ability of a prospective employee to a potential employer by sending her a signal. Based on this view which has been developed by Michael Spence and other scholars, credentials reveal the unobserved ability of the employees to the employers which aids the employers to distinguish low and high ability workers from each other. After the employee enters the job market and starts working, the role of ability in determining the wages increases because ability is more observable to the employer. Over time, the informational role of credentials in determining wages decreases because the employers can observe their employee’s abilities. According to this view, people with higher levels of productivity choose higher levels of education to signal their ability to potential employers. Arcidiacono et al. (2010) apply the Armed Forces Qualification Test (AFQT) as a measure of ability and find that the ability of high school graduates is revealed gradually and over time to the job market, but the ability of college graduates is observed almost perfectly in the job market. This finding of this study suggest that a college degree may serve a signaling purpose. According to the human capital view developed by Gary Becker and other scholars, investment in education enhances productivity and therefore wages. Becker argues that “Education, training, and health are the most important investments in human capital (Becker 1992, p. 1.).” The supporters of human capital view argue that credentials do not serve employees merely due to their informational value in sending signals to the employers, but people improve their human capital by attending school (Becker, 1964; Ben-Porath, 1967).

Similarly, I would like to see the authors temper their claim that eligibility for Social Security benefits had a positive impact on college enrollment. Given the absence of pre-trends and the potential of confounding, secular trends that explain decreases in college enrollment before and after the policy, the authors should refrain from using causal language to link social security benefits and college enrollment. Besides, making this causal claim is not necessary for their primary claim about whether education affects noncognitive skills or not. Indeed, I raised a similar comment in the previous round of reviews and the authors do not seem to have addressed it.  I’ve pasted my comment from the earlier review here:

“Thirdly, I’m skeptical that the change in college enrollment pre and post the Social Security policy shift is necessarily due to the policy shift. Ideally, a diff-in-diff would should secular trends for several years pre-policy change and then a jump. Given the data limitations of NLSY, this doesn’t seem to be possible. The authors also mention a Dynarski piece on page 6 that documents decreasing college enrollment in the years leading up to the policy shift. It’s possible that the enrollment difference pre- and post-policy are nothing more than these secular trends and have nothing to do with an exogenous shock from a policy change...”

In conclusion, the authors tackle a question that has scholarly significance. Their analysis will hardly be the last word on the topic. Between (1) the sensitivity of the results to models with and without baseline measures of noncognitive skills and (2) lingering doubts as to how exogenous a Social Security policy shock really is, readers would not unreasonable to see this study as not providing compelling evidence about education affecting noncognitive skills. There is nothing wrong to publishing a paper with these limitations, but an appropriate tone in which claims are not overstated and comport to the empirical results are warranted as others continue to study the topic.

We have modified the interpretations of table 3:

Previously we described the results as follows:

Table 3 represents the impact of eligibility for student benefits on years of schooling by age 23, complete college by age 23, and complete college by age 28. Note that schooling is not a binary variable, but it is the actual years of schooling. However, complete college by age 23 and 28 are binary variables. The results show positive impact of eligibility for student benefits on years of schooling and completing any years of college. That means that eligibility for student benefits is associated with almost half a year increase in schooling per person. Please see Appendix C for more details.

And now they are changed to the following:

Table 3 represents the impact of eligibility for student benefits on years of schooling by age 23, complete college by age 23, and complete college by age 28. Note that schooling is not a binary variable, but it is the actual years of schooling. However, complete college by age 23 and 28 are binary variables.  The results do not show significant impacts of eligibility for student benefits on years of schooling by age 23 and completing any years of college by age 23 when no control variable is included in the regressions. However, in most cases we see positive and significant impacts when the control variables are included. These results could be interpreted as positive impacts of the student benefits program on education outcomes. These findings, however, should be interpreted with caution because as mentioned above the impacts are not significant when control variables are not included. Another reason is that NLSY 79 dataset does not include data prior to 1979. This also, might affect the findings presented in Table 3. Please see Appendix C for more details.

In addition, we have modified the interpretations of tables 4 and 5:

Previously we described the results as follows:

The second row shows the impact of years of schooling completed by age 23 on the standardized Rosenberg Self-Esteem Scale measured in 2006. In 2006, the youngest participant in NLSY dataset is 42 years old. Hence, other than some rare exceptions, everybody should be out of school by 2006. Here, the coefficients are smaller compared to those in the first row and we still see some positive impacts of years of schooling on the Rosenberg Self-Esteem Scale measured in 2006 when the participants ages range from 42 to 49.

And now they are changed to the following:

The second row shows the impact of years of schooling completed by age 23 on the standardized Rosenberg Self-Esteem Scale measured in 2006. In 2006, the youngest participant in NLSY dataset is 42 years old. Hence, other than some rare exceptions, everybody should be out of school by 2006. Here, the coefficients are smaller compared to those in the first row and we still see some positive impacts of years of schooling on the Rosenberg Self-Esteem Scale measured in 2006 when the participants ages range from 42 to 49. Nevertheless, note that the coefficients are statistically significant only at 10 percent level in columns (1) to (5) and they are not statistically significant in column (6).

Also, on the same page we deleted the following sentence:

Moreover, our results assure us that the variation in aid eligibility has created a significant variation in college attendance and years of schooling.

From the following paragraph:

As mentioned before, the goal of this paper is to understand the impacts of schooling on non-cognitive abilities rather than the impacts of aid on schooling. Dynarski (2003) studied the impacts of variation in aid on years of schooling and college attendance. Here, not only we replicated her results and found similar impacts of aid on education, but we provided more details on the impacts of aid eligibility on schooling. Our findings to this point give us confidence that we have estimated the impacts of eligibility for Social Security student benefits on schooling correctly. Moreover, our results assure us that the variation in aid eligibility has created a significant variation in college attendance and years of schooling. We apply this variation in our 2SLS specifications to estimate the impacts of years of education on non-cognitive abilities. However, before discussing the 2SLS results we have presented the results of least-square estimations.

Further, on the same page we deleted the following sentence:

The impacts of years of schooling by age 23 on the Standardized Rotter Locus of Control Scale varies between 7.2 and 9.5 percentage points.

From the following paragraph:

The last row in Table 4 represents the impacts of schooling by age 23 on the Standardized Rotter Locus of Control Scale measured in 2014. As can be seen, the coefficients in all three rows are positive, which means more years of schooling is associated with an increase in the sense of control over life (positive impacts on internal locus of control and negative impacts on external locus of control). The impacts of years of schooling by age 23 on the Standardized Rotter Locus of Control Scale varies between 7.2 and 9.5 percentage points.

Also, we have modified the interpretations of tables 6 and 7:

Previously we described the results as follows:

Perhaps the most important aspect of the results in Tables 6 and 7 is that education has a significant impact on non-cognitive skills when we do not control for a previous level of non-cognitive skill, but when the previous levels of non-cognitive skills are controlled for we hardly see any significant impact. In Tables 6 and 7 when a previous level of self-esteem is not included as a control variable, we can see a significant and positive impact of education on self-esteem and when a previous measure of self-esteem is included as a control variable, the impacts are still positive, but they are either are not statistically significant or they are significant at 10 percent level. The impact of education on locus of control is positive and statistically significant at 10 percent level when a previous level of locus of control is not controlled for, but when it is added as control variable the impact is not statistically significant, but the coefficients stay positive. That means the impact of education is such that it shifts locus of control from external to internal levels, but the impact is hardly statistically significant.

And now it is changed to the following:

Perhaps the most important aspect of the results in Tables 6 and 7 is that education has a significant impact on non-cognitive skills when we do not control for a previous level of non-cognitive skill, but when the prior levels of non-cognitive skills are controlled for we hardly see any significant impact. In Tables 6 and 7 when a previous level of self-esteem is not included as a control variable, we can see a significant and positive impact of education on self-esteem and when a previous measure of self-esteem is included as a control variable, the impacts are still positive, but they either are not statistically significant or they are significant only at 10 percent level. The impact of education on locus of control is positive and statistically significant at 10 percent level when a previous level of locus of control is not controlled for, but when it is controlled for then the impact is not statistically significant.

Conclusion of the paper:

The following sentence:

Our difference-in-differences results show a positive impact of eligibility for student benefits on years of schooling and completing any years of education at college.

Is changed to the following:

Our difference-in-differences results suggest a positive impact of eligibility for student benefits on years of schooling and completing any years of education at college, but as mentioned before these findings should be interpreted with caution do to limitations of NLSY79 dataset. 

The second half of following paragraph:

Note that the results of this paper do not entirely deny the positive impacts of formal college education on non-cognitive abilities. In fact, we also find positive impacts, but when we control for the previous levels of non-cognitive skills, we rarely see any significant impact. Hence our results are in line with the findings of Wolfle and Robertshaw (1982) and Pascarella and Terenzini (1991) who found similar results decades ago. As mentioned before, several scholars report positive impacts of education on non-cognitive skills which is a different finding from that of Wolfle and Robertshaw (1982) and Pascarella and Terenzini (1991), but our results do not confirm any causality relationship from college education to non-cognitive skills. The results of this paper confirm that the Social Security student aid program had positive and significant impacts on educational outcomes, including length of education and college attendance. However, despite the fact that the impacts of education on non-cognitive skills are positive, but they are not significantly different from zero.

Is changed to the following:

Note that the results of this paper do not entirely deny the positive impacts of formal college education on non-cognitive abilities. In fact, we also find positive impacts, but when we control for the previous levels of non-cognitive skills, we rarely see any significant impact. Hence our results are in line with the findings of Wolfle and Robertshaw (1982) and Pascarella and Terenzini (1991) who found similar results decades ago. As mentioned before, several scholars report positive impacts of education on non-cognitive skills which is a different finding from that of Wolfle and Robertshaw (1982) and Pascarella and Terenzini (1991), Our results do not confirm any causality relationship from college education to non-cognitive skills. The results of this paper suggest that perhaps the Social Security student aid program had positive impacts on educational outcomes, including length of education and college attendance. However, despite the fact that the impacts of education on non-cognitive skills are positive, but they are not significantly different from zero.

New references:

Spence, M. (1978). Job market signaling. In Uncertainty in economics (pp. 281-306). Academic Press.

Arcidiacono, P., Bayer, P., & Hizmo, A. (2010). Beyond signaling and human capital: Education and the revelation of ability. American Economic Journal: Applied Economics2(4), 76-104.

Becker, G. S. (1992). Human capital and the economy. Proceedings of the American philosophical society136(1), 85-92.

Ben-Porath, Y. (1967). The production of human capital and the life cycle of earnings. Journal of political economy75(4, Part 1), 352-365.

Becker, G. S. (1964). Human capital. A Theoretical and Empirical Analysis with Special References to Education.